# Distilling an End-to-End Voice Assistant Without Instruction Training Data

## Abstract

Voice assistants, such as Siri and Google Assistant, typically model audio and text separately, resulting in lost speech information and increased complexity. Recent efforts to address this with end-to-end Speech Large Language Models (LLMs) trained with supervised finetuning (SFT) have led to models "forgetting" capabilities from text-only LLMs. Our work proposes an alternative paradigm for training Speech LLMs without instruction data, using the response of a text-only LLM to transcripts as self-supervision. Importantly, this process can be performed without annotated responses. We show that our Distilled Voice Assistant (DiVA) generalizes to Spoken Question Answering, Classification, and Translation. Furthermore, we show that DiVA better meets user preferences, achieving a 72% win rate compared with state-of-the-art models like Qwen 2 Audio, despite using >100x less training compute.

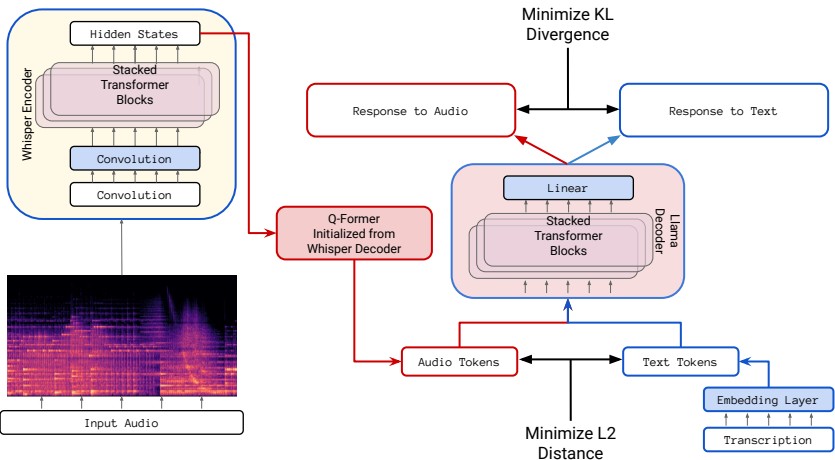

Figure 1: Training Pipeline for Distilled Voice Assistant (DiVA), Red indicates trainable components while Blue indicates frozen pretrained modules. DiVA modifies a text-only LLM into a general purpose Speech LLM by using the model's own responses to transcribed speech as self-supervision.

## 1 Introduction

As Large Language Models (LLMs) capabilities increase, so does the value of bringing these capabilities to new modalities, including audio and speech (Shu et al., 2023; Wang et al., 2023; Gong et al., 2023). Speech is a natural interaction surface for language technology (Murad et al., 2019), offering measurable efficiency gains for users (Ruan et al., 2018). One straightforward method of integrating speech with LLMs is to feed audio to an Automatic Speech Recognition (ASR) model and produce a text transcription for the LLM to use. However, this process loses meaningful information carried through tone, pacing, and accent (Upadhyay et al., 2023) regardless of the transcription accuracy. Furthermore, finetuning these pipelined systems requires supervision for both transcription and response generation, increasing annotation complexity and costs.

Table 1: High-Level comparison with state-of-the-art open-access Speech & Audio LLMs which we compare to. DiVA offers an entirely different approach to training using context distillation.

| Model | Base LLM | Training Method | # Tasks | # Hours |
|---|---|---|---|---|
| SALMONN | Alpaca | SFT | 12 | 4400 |
| Qwen Audio Chat | Qwen Chat | SFT | 31 | ~50k |
| Qwen 2 Audio Instruct | Qwen2 Instruct | SFT, DPO | Unreported | >370k |
| DiVA (Ours) | Llama 3 | Distillation | N/A | 3.5k |

As such, LLMs that interface with speech directly have the potential to accelerate inference, reduce annotation costs, and capture the rich information inevitably lost by ASR. In this pursuit, a variety of works have trained audio-encoders on top of LLMs (Ao et al., 2021; Chen et al., 2021b; Deshmukh et al., 2023; Chu et al., 2023; Wu et al., 2023a), many of which utilize the same well-established approach: large-scale multi-task supervised finetuning (SFT).

Models using SFT face several challenges. First, without a large degree of task diversity in their training data, they often fail to generalize capabilities from the text-only LLM to speech. As observed in Tang et al. (2023), freezing the weights of the text-only LLM is insufficient to prevent this "forgetting". In order to generalize well, SFT must be trained with labeled data from a wide range of tasks and domains, with minimal imbalance between tasks. However, broad annotated speech instruction training data does not currently exist.

The limited instruction data that does exist is often collected from a small pool of speakers (Kim et al., 2021; Tomasello et al., 2023) or intended for evaluation rather than training (Faisal et al., 2021; Eisenstein et al., 2023). This lack of representation of speech from the wider population is likely to exacerbate biases in speech processing (Koenecke et al., 2020; Mengesha et al., 2021; Chan et al., 2022; Javed et al., 2023; Brewer et al., 2023). At present, Speech LLMs appear fundamentally limited by existing instruction data.

In this work, however, we argue that these "limitations" of existing data are artificially imposed by SFT. The speech community has already invested in large-scale data collection from the internet (Radford et al., 2023; Chen et al., 2021a; Li et al., 2023b), audiobooks (Panayotov et al., 2015; Pratap et al., 2020), and public archives (Galvez et al., 2021). Furthermore, several datasets have been explicitly gathered to represent diverse demographics (Porgali et al., 2023; Garg et al., 2023). However, these large-scale and diverse datasets are dominated by data in just one task: Automatic Speech Recognition (ASR). This means that models trained with SFT cannot make use of the entirety of this data without "forgetting" non-ASR capabilities.

We solve the "forgetting" problem by training a model that generalizes well despite using *only* ASR data. Rather than relying on external labels, our **Di**stilled **V**oice **A**ssistant (**DiVA**) self-supervises learning using the output distribution of an LLM in response to transcripts as a target, a cross-modal form of context distillation (Snell et al., 2022; Mu et al., 2024). We test our approach by training on just a single corpus, the CommonVoice, consisting of speech and transcriptions contributed by volunteers around the world and recorded on their own devices (Ardila et al., 2019).

Despite this data simplicity, DiVA generalizes to Spoken Question Answering, Classification, and Translation. Furthermore, DiVA is preferred by users to our most competitive baseline Qwen 2 Audio in 72% of trials despite DiVA using over 100x less training compute. Beyond contributing a new Speech LLM, DiVA creates a new approach to Speech LLMs that trains more efficiently and generalizes better *without* requiring investment in new speech instruction data.

## 2 RELATED WORK

LLMs have been extended to both audio and image inputs using cross-modal encoders. For example, LLaVA (Liu et al., 2023b) enables image understanding by connecting CLIP (Radford et al., 2021) to Vicuna (Chiang et al., 2023) through an MLP layer. Several recent works (Zhang et al., 2023; Gong et al., 2023; Tang et al., 2023; Chu et al., 2023; 2024) have connected audio-encoders (Gong et al., 2021; Hsu et al., 2021) to LLMs. There are two critical questions in this space.

**How can audio features be transformed into a reasonable number of LLM input embeddings?** Audio comes at high sample rates, and therefore, audio encoders often have a large number of outputs. For these features to be usable by the LLM, the dimensionality must be reduced, either by stacking consecutive features (Wu et al., 2023a; Fathullah et al., 2024) or learning an adapter-module, such as an MLP (Liu et al., 2023b; Gong et al., 2023), or Q-Former (Dai et al., 2023; Tang et al., 2023).

While learned approaches are more flexible, allowing for an adaptive reduction, they generally require learning a cross-attention mechanism, which generally requires significant training (Li et al., 2023a). In this work, we find the best of both worlds by leveraging the Whisper decoder (Radford et al., 2023) to initialize the text-audio cross-attention mechanism of a Q-Former.

**How can speech LLMs be trained to achieve instruction following abilities using existing data?** Prior work has explored two routes for creating instruction data without major financial investment. The first approach is to transform many existing datasets into an instruction-following format (Dai et al., 2023; Chu et al., 2023; Tang et al., 2023; Dai et al., 2023; Liu et al., 2023a). In this case, limitations are often set by datasets that are not aligned with intended LLM usage and imbalances across tasks. The second approach is to train on synthetic responses to text representations of the new modality from commercial models (Liu et al., 2023b; Gong et al., 2023).

Our approach is most related to the latter. Rather than generating data with an external model, we capitalize on the idea that instruction-tuned language models can provide valuable learning signals as long as the input is within the textual modality. Using the output distribution in response to text transcripts, we can more strongly guarantee the transfer of existing capabilities using context distillation (Snell et al., 2022; Mu et al., 2024).

**How can we train foundation models for speech using open and permissively licensed data?** Recently, frontier LLMs have begun integrating native speech capabilities. Unlike prior speech foundation models (Baevski et al., 2020; Hsu et al., 2021; Chen et al., 2022; Kim et al., 2021; Peng et al., 2023; 2024), these models offer virtual assistant capabilities rather than self-supervised audio representations or transcriptions. It is unclear to what degree these results are dependent on internal datasets, especially since even the state-of-the-art *open-access* Speech LLM with such capabilities reports no training data details other than size (Chu et al., 2024).

Similar to the Open Whisper-style Speech Model (OWSM) initiative (Peng et al., 2024), we use only open and permissively-licensed data. Furthermore, unlike the baselines we compare to in Table 1, we release the training code, rather than just the inference code, which can help reproduce a DiVA-style model easily. Beyond our novel method, we believe this too broadens the ability to train and understand Speech LLMs.

## 3 METHOD

DiVA is an end-to-end voice and text assistant, trained using the process shown in Figure 1. We focus heavily on effectively using pretrained models in each domain. Similar to prior works, we initialize the audio encoder from the 1.5B parameter Whisper-Large-v3 model. Unlike prior works, we use all components of Whisper: not only reusing the encoder but also initializing a Q-Former from the decoder. We train this architecture using distillation loss on the input and output distribution of the text-only LLM, which we discuss in Section 3.2.

### 3.1 MODEL INITIALIZATION

When adding multimodal capabilities to an existing language model, the new modality must be represented as embeddings that can used in place of text token embeddings. Achieving this goal has two steps. First, meaningful features must be extracted from the input modality. Second, these features must be aggregated to be in-distribution for the downstream language model.

**Audio Feature Extraction** We follow prior works (Chu et al., 2023; Tang et al., 2023) and use the Whisper encoder (Radford et al., 2023). Whisper first transforms the raw audio signal into a 128-channel time-frequency domain Mel-spectrogram. This is then passed through two 1D convolutions and used as embeddings fed to an unmodified Transformer architecture (Vaswani et al., 2023).

**Audio-Text Feature Alignment** While the Whisper encoder extracts meaningful audio features, they are encoded at high granularity, with one token for every 40 milliseconds of input audio. By comparison, humans speak around one syllable every 150 milliseconds on average across languages (Coupé et al., 2019), and most tokens in an LLM vocabulary are made up of several syllables. This creates a mismatch between the granulatity between the Whisper encoder outputs and the downstream LLMs input distribution.

Prior work (Tang et al., 2023) addresses this using a Querying Transformer (Q-Former, Li et al. 2023a), which learns static query embeddings with cross-attention to keys and values features from another modality. Given audio embeddings $\boldsymbol{A}$, the Q-Former learns a transformer with a cross attention mechanism $\sigma(\frac{\boldsymbol{Q}(\boldsymbol{K}\boldsymbol{A}^{\mathsf{T}})}{\sqrt{d_k}})(\boldsymbol{V}\boldsymbol{A})$ where $\boldsymbol{Q}$ is a static set of query vectors, while $\boldsymbol{K}$ and $\boldsymbol{V}$ are projection matrices for the audio tokens. Conceptually, this cross-attention mechanism learns to dynamically aggregate information from the audio tokens into text-like tokens. This comes at the cost of significant training required to train the transformer from scratch.

The Whisper decoder, which prior work discards, is trained with a similar goal for ASR: mapping audio embeddings to discrete text tokens. Therefore, rather than learning Q-Former parameters from scratch, we initialize $K$ and $V$ from Whisper's cross-attention mechanism. We adapt the model to a Q-Former by replacing the inputs with static query tokens $Q$. Finally, we project the output from the hidden dimension $h$ of Whisper to the hidden dimension $H$ of the LLM. This results in a set of $\{\boldsymbol{t}_q^{audio} \in \mathbb{R}^{H \times |Q|}\}$ output tokens representing the audio.

**Text Decoding** For language processing and instruction following capabilities, we use Llama 3 (Dubey et al., 2024)[1] and leave its weight frozen throughout training.

### 3.2 DISTILLATION LOSSES

We optimize two loss functions based on audio recordings and corresponding text transcripts from ASR data. First, we minimize the distance between embeddings of audio and text on the *input* side of the LLM, similar to prior work for Vision-LMS (Radford et al., 2021; Li et al., 2023a). Then, we minimize the KL Divergence between the *output* distribution in response to audio and text as a form of cross-modal context distillation (Mu et al., 2024; Snell et al., 2022).

#### 3.2.1 CROSS-MODAL TOKEN ALIGNMENT LOSS

To capture the mutual information between recordings and text transcripts, for a given ASR example (a text transcript and an audio recording), we align speech and text tokens as follows: The text transcript is embedded as $N$ text tokens $\boldsymbol{t}_i^{text} \in \mathbb{R}^{H \times N}$. The model produces $|Q|$ tokens from the audio recording where $|Q| > N$. We align these representations by minimizing the $L_2$ distance between the text embeddings and the final $i$ audio embeddings:

$$L_{con} = \sum_{n=0}^{N} |\boldsymbol{t}_n^{text} - \boldsymbol{t}_{Q-N+n}^{audio}|_2 \tag{1}$$

We use the final $N$ tokens of the audio embedding rather than the initial $N$ tokens due to the causal attention in Whisper's decoder. Since the final tokens can attend to all preceding tokens, aligning the representations of the final tokens backpropagates signal to every token in the sequence. On the other hand, the additional $Q - N$ tokens provide information bandwidth for other information, such as sociophonetic cues, to be passed to the LLM.

Empirically, as we explore in Section 6, training with only token alignment leads to poor model quality, even when low loss is achieved. However, token alignment appears to enable reasoning between text and audio tokens, vastly improving text instruction adherence.

---

[1]Training was performed before the release of Llama 3.1.

### 3.2.2 Distillation from Output Embedding Distance

Voice assistant models should give coherent, helpful, and harmless responses to user speech. Thankfully, many openly accessible text-only LLMs have been extensively refined for these objectives. As such, our challenge is not to learn these behaviors but instead to transfer them to the audio modality. While, in theory, input token alignment could achieve this, even minor differences in input embeddings can significantly affect model behavior in practice (Cai et al., 2022).

Distillation loss, on the other hand, directly optimizes for the similarity of the output distribution (Hinton et al., 2015). Rather than distilling a large model into a smaller model, recent work has applied to distilling useful context into model weights, a process termed context distillation (Snell et al., 2022; Mu et al., 2024). Here, we apply context distillation across modalities, aiming to distill a text context into the audio modality under the assumption that the model should respond similarly to audio and text for most inputs.

In prior context distillation works, the full Kullback–Leibler (KL) Divergence has been shown to be prohibitively expensive at training time due to the large vocabulary of modern LLMs. Therefore, the KL Divergence is instead approximated by sampling random tokens (Snell et al., 2022). In our case, where the output embedding matrix is frozen, we show that there is an objective function easier to optimize:

**Lemma 1.** *Given the probability $P_t$ from a teacher model and the probability $P_s$ from a student model, the KL Divergence is defined as $KL(P_t, P_s) = P_t \cdot (\log P_t - \log P_s)$. For a transformer language model, $P_s = \sigma(O_s h_s)$ where $h_s$ is the final hidden state, $O_s$ is the output embedding matrix, and $\sigma$ is the softmax function. Let $\theta_s$ be the student weights which we are trying to train to minimize the KL Divergence, then*

$$\arg_{\theta_s} \min ||\boldsymbol{h}_s - \boldsymbol{h}_t||_2 \subset \arg_{\theta_s} \min KL(P_t, P_s)$$

*Proof.* The KL divergence is minimized when $P_s = P_t$. Based on our definition of LM probability, this is equivalent to achieving $\sigma(O_s h_s) = \sigma(O_t h_t)$. In the special case we consider, where the teacher and student are initialized from the same weights, and $O_s$ is held constant, we know that $O_s = O_t$. Thus, a non-unique global minimum will be achieved when $h_s = h_t$, where the non-uniqueness comes from the softmax function $\sigma$, which is not injective. □

More importantly, we find that: (1) The gradient for L2 loss is much smoother than minimizing the KL divergence empirically[2]. (2) Since the vocabulary size of most modern LLMs is far larger than the hidden dimension, the distance between hidden states can be computed using far fewer operations than the KL divergence. In practice, we optimize the similarity of only the first predicted next token (after all $I$ text tokens/all $Q$ audio tokens) for efficiency, as Morris et al. (2023) has shown that just a single token probability encodes significant information, both for prior and future tokens.

Notably, training with this loss only guarantees that the output distribution is well aligned in response to audio. However, our intuition is that this loss alone is likely to be less robust to input distribution shift without our token alignment loss, which we explore in Section 6.

## 4 Experimental Setup

### 4.1 Training Data

We utilize the English subsection of CommonVoice 17 (Ardila et al., 2019) as the dataset for all DiVA training runs. The dataset comprises just over 3.5 thousand hours of read text that has been crowdsourced and validated on the CommonVoice website. We select the CommonVoice for three reasons. Firstly, it is permissively licensed for commercial and research use. Secondly, it contains speech recorded in realistic settings on an individual's device rather than in a professional studio. Finally, it includes speech from 93,725 speakers from a global pool of volunteers[3]. The first factor means that the resulting DiVA models we release can be adopted for use broadly, while the latter two help make the training data more representative of real users.

---

[2]We explore this with an isolated small-scale experiment in Appendix A.2

[3]Statistics drawn from the official CommonVoice tracker

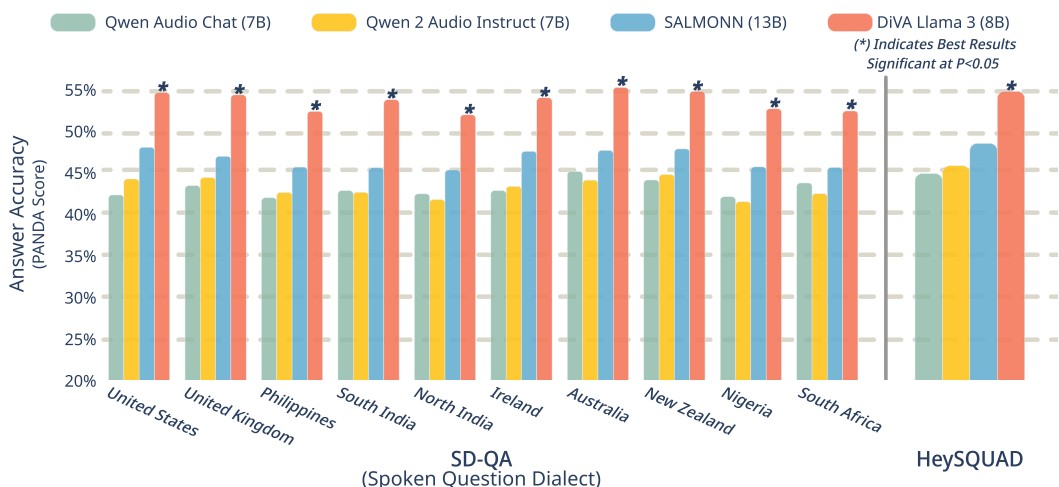

Figure 2: Results across our two Question Answering benchmarks covering both standard evaluation and robustness to regional accents. Model correctness is assessed using the PANDA metric, which is tuned for strong correlation with human judgments of correctness (Li et al., 2024), and significance is from a paired bootstrap test (Dror et al., 2018).

## 4.2 TRAINING HYPERPARAMETERS

We train for 4300 steps and a batch size of 512 using the AdamW Optimizer, a learning rate of 5E-5, and a weight decay of 0.1. This amounts to roughly two epochs over the data. We linearly warm up the learning rate for the first 1% of steps and then follow a cosine learning rate learning rate schedule which decays the learning rate to 0 over the course of the training run. The training run completes in approximately 12 hours on on a TPU v4-256 pod.

## 5 QUANTITATIVE AND QUALITATIVE EVALUATIONS

We first assess how DiVA compares to baseline models for various spoken language benchmarks SFT models target. We evaluate on benchmarks for spoken question answering, speech classification, and speech translation. This provides a quantitative validation of DiVA's generalization.

However, these benchmarks were all designed to test single task systems focused on each individual task. It is unclear whether these benchmarks capture the capabilities users expect from virtual assistants that speech LLMs are now powering commercially. To assess this, we run a further side-by-side comparison of DiVA with the best performing model on the benchmark evaluation Qwen 2 Audio (Chu et al., 2024).

**Baselines** We compare our results to three openly available Speech Language Models: SALMONN, Qwen Audio Chat, and Qwen 2 Audio Instruct. Notably, all the baseline models utilize SFT covering these benchmark tasks. This makes them strong baselines: they all use similar scale base LLMs to DiVA, all make use of the Whisper encoder, and all have received direct supervision on the evaluated tasks. For our user study, we compare with Qwen 2 Audio, which reports state-of-the-art numbers and achieves the best average performance in our benchmarks.

## 5.1 BENCHMARKING

For question answering, we use HeySquad (Wu et al., 2024) and SDQA (Faisal et al., 2021), testing on 4,000 and 494 question-answer pairs, respectively. Classification is broken down into emotion recognition, sarcasm detection, and humor recognition. Emotion recognition is assessed on IEMO-CAP (Busso et al., 2008) and MELD (Poria et al., 2019), with 1,241 and 2,608 utterances. We evaluate sarcasm detection on MUSTARD's 690 clips (Castro et al., 2019) and humor recognition on URFunnyV2's 7,614 examples. Finally, speech translation is tested on CoVoST 2, translating

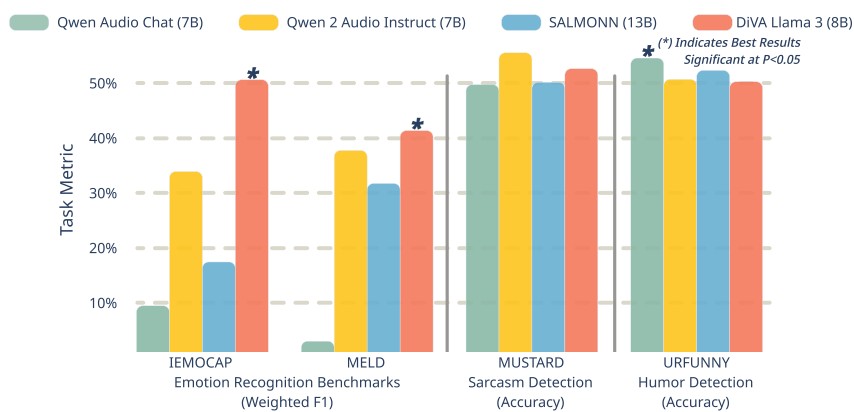

Figure 3: Results across Emotion, Humor, and Sarcasm classification tasks. We measure class-weighted F1 for multi-class classification and accuracy for binary classification. Significance computed using a paired bootstrap test.

15,500 English examples into seven commonly-tested typologically diverse languages (Clark et al., 2020). These datasets cover a wide range of traditionally benchmarked speech tasks drawn from prior work, which we cover in greater depth in Appendix A.3.

### 5.1.1 SPOKEN QUESTION ANSWERING

We evaluate all models on zero-shot spoken question answering by prompting them with recorded audio of a speaker asking a question and the prompt: *You are a helpful assistant. Give answers as a simple single sentence*. The underlying LLMs for all baseline models are capable of question-answering, meaning that the audio encoder only needs to learn to map audio to the correct corresponding text to achieve strong results. This is a case where we expect DiVA to perform particularly well despite never having been explicitly trained on spoken questions.

Empirically, this expectation is met as shown in Figure 2. DiVA significantly (P<0.05) over the baselines by at least 10% (+5 PANDA) across both benchmarks and all accents.

However, it's unclear whether lower accuracy can be directly attributable to "forgetting". We qualitatively explore this question by labeling a sample of 50 responses from the HeySQUAD dataset for whether the responses include even an attempted answer relevant to the task. Qwen Audio shows signs of severe forgetting, with 30% of responses ignoring the prompt instructions entirely and instead transcribing the question e.g. *"The citation for the Pearson v. Society of Sisters case is What is the citation for the Pearson v. Society of Sisters case?"*. By comparison, SALMONN, which takes inference time interventions to reduce overfitting by partially ablating the LoRA modules learned for the base LLM, sees reduced overfitting with only 8% of model responses ignoring the prompt and instead transcribing. Qwen 2 Audio sees further reduced overfitting, likely due to its DPO process using unreleased data, with only 4% instances where the instruction is ignored. DiVA, despite being trained only on transcription data, is the only model adheres to the instruction consistently.

### 5.1.2 SPEECH CLASSIFICATION

One possible downside of our distillation approach is that the loss function contains minimal supervision for tasks where the audio of speech itself contains rich information through tone. However, tone is frequently correlated with the semantics of the text itself. We hypothesize this may enable the audio encoder to transfer some amount of this tone information based on weak supervision from text. To assess this, we evaluate on speech classification tasks where tone is likely to play a major role: Sarcasm Detection, Humor Detection, and Emotion Recognition.

**Emotion Recognition** For emotion classification, we prompt each model with the instructions *Respond in a single word what emotion the input exhibits. If there is no clear emotion, respond*

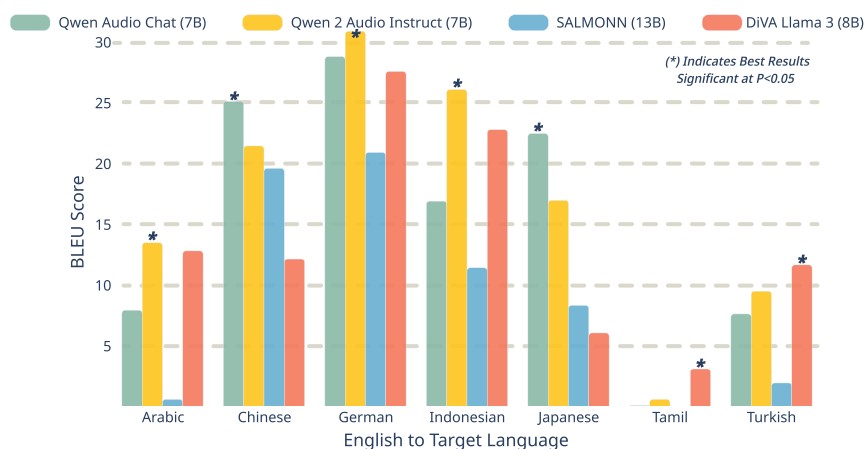

Figure 4: Results for Speech Translation across 7 typologically diverse languages. We evaluate using SacreBLEU and compute confidence intervals using a Paired Bootstrap.

*'Neutral'*. To use each model as a classifier, we follow prior work (Hendrycks et al., 2021) and use the log-probabilities assigned to each possible label as classifier scores.

DiVA performs significantly better than both baselines on both the MELD benchmark, sourced from television, and IEMOCAPS, which operates over recorded conversations. In comparison to DiVA, both baseline models struggle to predict a diverse array of labels. Qwen Audio predicts the emotion as Sadness for greater than 90% of inputs for both MELD and IEMOCAPS, while SALMONN and Qwen 2 Audio behaves similar with Neutral predictions.

These results are quite surprising given that DiVA is trained without explicit emotion supervision. However, many examples in both IEMOCAPS and MELD communicate emotion through both text and audio signals which may confound how well these evaluations capture true sociophonetic signal.

**Sarcasm & Humor Detection** We also evaluate on two tasks where communicative intent is expressed largely through tone. For Sarcasm Detection, we prompt each model to *Respond whether the input is sarcastic. Answer with a simple yes or no* for sarcasm detection and *Respond whether the input is intended to be humorous. Answer with a simple yes or no*. In both cases, we compare the log-probability assigned to either the token "Yes" or the token "No.

No model performs particularly well in these tasks. None of the evaluated models perform significantly ($P > 0.05$) better than chance on sarcasm detection and only Qwen Audio Chat performs better than chance on Humor Detection. This suggests there is significant progress to be made in enabling speech-oriented language models to understand more complex social signals in speech.

These tasks also highlight a shortcoming of distillation – namely, that DiVA inherits even non-desirable behaviors from the base LLM. Even when asked whether obviously very serious text is humorous or sarcastic, Llama 3 will almost always find a way to argue that it is humorous. DiVA inherits this behavior, predicting the "Yes" label in both tasks over 90% of the time.

### 5.1.3 SPEECH TRANSLATION

Finally, we assess the speech-to-text translation capabilities of each model from English Speech to text in another language. We prompt each model with the instruction *Translate the input from {input_lang} to {output_lang}*.

Results on this benchmark are far more mixed. The original Qwen Audio performs best on Chinese and Japanese, Qwen 2 Audio performs best on Arabic, German and Indonesian, and DiVA performs best on Tamil and Turkish. Notably, the original Qwen trains with more than 3700 hours of speech-to-text translation data from CoVost2. While Qwen 2 does not report which tasks it trains on, it is likely it trains on similar or increased volumes of data from CoVost2 as the original Qwen. This

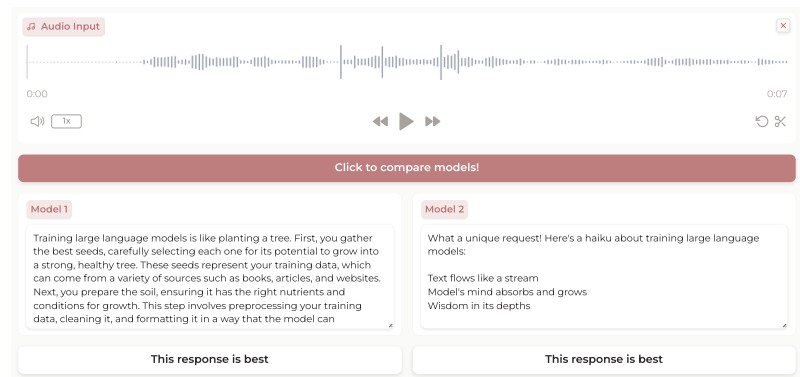

Figure 5: Example of the double-blind interface for the user study with responses (Left: Qwen 2, Right: DiVA) to the speech *Can you tell me about Large Language Models in the style of a haiku?*.

| Qwen 2 Audio | DiVA | Difference |
|---|---|---|
| 28% Win Rate | 72% Win Rate | Significant at |
| (148 Votes) | (374 Votes) | P<0.001 |

Figure 6: Win-rate between models in our 522 preferences from 53 Prolific users.

highlights the data and compute efficient transfer of the DiVA approach, as both of these models trained on more translation specific data than DiVA used for it's entire training.

DiVA's most notable underperformance is in Chinese and Japanese, where it underperforms both other models. Inspecting DiVA's outputs and comparing them to translations from Llama 3 in response to text, we again find that our distillation loss leads us to preserve a negative behavior — for both Chinese and Japanese, Llama 3 has a strong bias towards generating translations in the Latin alphabet (Pinyin and Romanji) rather than the expected native script. This leads to especially poor results in these languages.

## 5.2 QUALITATIVE USER STUDY

Finally, to get a sense of how well suited the resulting models are to user preferences, which may or may not be well captured in existing benchmarks, we recruit participants to compare DiVA to the top performing baseline, Qwen 2 Audio.

### 5.2.1 RECRUITMENT & STUDY DESIGN

We recruit 53 participants on the Prolific platform to provide preference ratings. Each user was allowed to contribute a maximum of 10 ratings, but able to opt-out at any time, resulting in 522 preference ratings comparing the models. We paid users 2.50$ per 10 ratings, which took fewer than 10 minutes of active time for all annotators involved, for an effective pay rate of 15$ per hour. We report annotator demographics in Appendix A.5.

We pre-screened for users who report familiarity with existing LLM chatbots and virtual assistants (e.g. ChatGPT, Gemini, Claude and others). In order to avoid biasing our participants, we prompt them without reference to specific tasks to *Record something you'd say to an AI Assistant! Think about what you usually use Siri, Google Assistant, or ChatGPT for.* Users were then shown responses from each model, without knowledge of which model was which. To avoid any positional bias, we shuffle the order which users were shown model responses for each recording submitted.

### 5.2.2 RESULTS

Despite no clear winner on all benchmarks for Qwen 2 and DiVA, DiVA generally is strongly preferred by users, with a 72% win rate at the preference level. At the user level, 41/53 (77%) of users preferred DiVA for the majority of their inputs. Beyond showing that DiVA improves preference alignment, this indicates that benchmarks may not correlate with practical usage.

# 6 LOSS ABLATION

To better understand each component of our distillation loss, we investigate the influence of each loss component independently. In Figure 7, we compare results between the complete DiVA method, using just the output distillation loss, and using just the input token alignment loss.

**Impacts of KL Divergence Loss** The most clear necessity for DiVA is the KL Divergence loss on the output distribution. Using token-alignment only does not simply lead to marginally worse results, it causes generations to be often incoherent. For generative tasks, the model often outputs sentences which are only vaguely semantically related to the input or unrelated markdown headers. In classification tasks, the token-alignment only model never performs significantly better than random guessing.

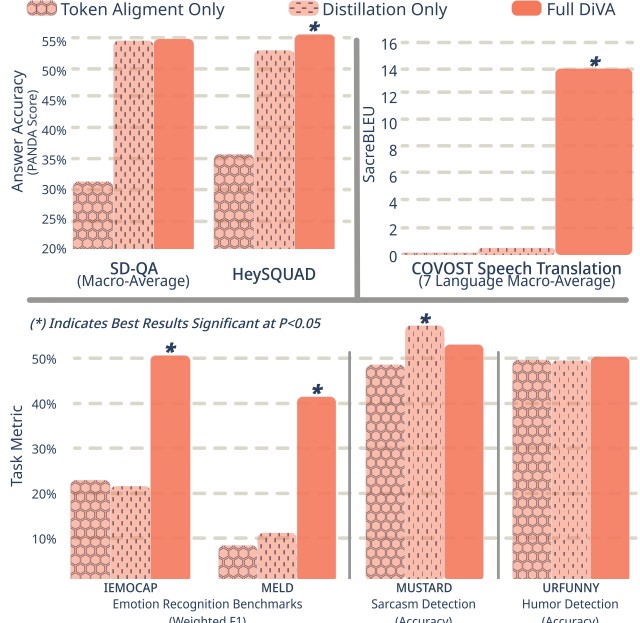

Figure 7: Ablation of the loss components from Section 3. Distillation leads to a capable audio-only model, but token-alignment improves instruction adherence.

**Impacts of Token Alignment Loss** This might raise the question: why use the token-alignment loss if it performs so poorly? In evaluations on question answering, this is certainly reasonable since using the KL Divergence loss alone already leads to stronger performance than the SFT baselines.

However, for translation and emotion recognition tasks, we see near-zero results from KL Divergence loss alone. Qualitatively, we observe that the distillation only model replies directly to the speech regardless of the text instructions.

We quantify this failure to adhere to instructions for the translation task using FastText Language ID (Joulin et al., 2017) on the outputs, under the assumption that outputs which are not in the correct target language are the result of ignored instructions. DiVA outputs the correct language 74% of the time while the distillation only model outputs the correct language only 1.4% of the time[4].

This indicates that the token-alignment loss is key to achieving a model using distillation that can follow both audio instructions and text instructions such as a system prompt.

# 7 CONCLUSION

In summary, we release DiVA, an end-to-end Voice Assistant model capable of processing text and audio natively. Our cross-modal distillation loss from text to speech showcases a promising direction for cost-effective capabilities transfer from one modality to another. Our Distilled Voice Assistant generalizes to Spoken Question Answering, Classification, and Translation despite only being trained on transcription data. Furthermore, DiVA is preferred by users to our most competitive baseline Qwen 2 Audio in 72% of instances despite DiVA taking over 100x less training compute. Together, these contributions highlight a path forward for rapid adaptation of LLMs to Speech, without large investments in new training datasets[5].

---

[4]We include LID results for all models in Appendix A.4

[5]We include anonymized links to training and evaluation code in Appendix A.1

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

# A APPENDIX

## A.1 REPRODUCIBILITY STATEMENT

We release our training code, as well as evaluation code, demo code & raw outputs at Anonymized links for the purpose of review. All dataset processing details are included in Appendix A.3.

## A.2 TOY EXPERIMENT ON KL DIVERGENCE VERSUS HIDDEN STATE ALIGNMENT

Beyond being a valid and efficient approximate of the KL Divergence, the $L_2$ loss should offer a more stable gradient, especially early in training when the output distributions are extremely different. When $P_t$ is positive and $P_s$ is near zero, the KL divergence explodes to extremely large values which can make optimization difficult and subject to significant numerical error.

In order to test this intuition, we set up a toy experiment where the student model outputs a single hidden state $h_s$ and the teacher model outputs a single hidden state $h_t$. In this highly simplified space, each model is fully parameterized by the these hidden states. We initialize and output vocabulary from the normal distribution with $32,000$ vocabulary items. Then, we optimize $h_s$ based on either the $L_2$ distance with $h_t$ or the KL Divergence with the output probabilities. Finally, for both procedures, we optimize for 100 steps with stochastic gradient descent, running the experiment 100 times at logarithmically increasing embedding dimensions, and plot the final KL divergence achieved under each loss function.

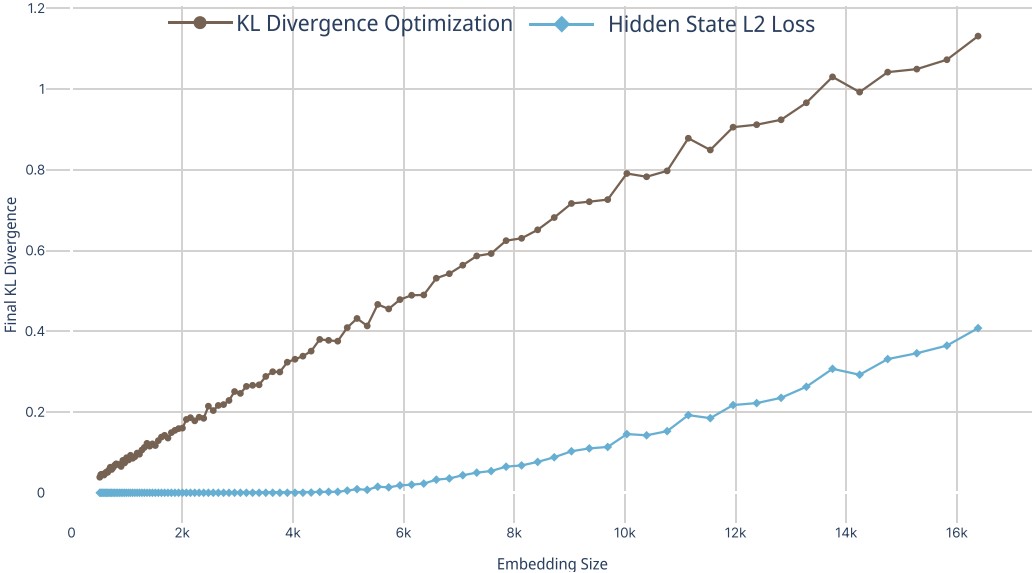

Figure 8: Empirical Comparison of the KL Divergence with our Proxy $L_2$ loss in a toy experimental setup. Optimizing the KL Divergence directly leads to *worse* KL Divergence than optimizing the $L_2$ loss. This gap increases as the hidden dimension becomes larger.

We see that, as the embedding dimension grows, optimizing the $L_2$ loss actually achieves *lower* KL divergence in this setup than optimizing the KL Divergence directly. To some extent, this makes sense as the $L_2$ loss is an incredibly simple convex function to optimize in this setting, while the KL divergence introduces significant additional complexity and a much sharper loss landscape early in optimization. We used this setup early in model design phases to help validate the choice of this approximation empirically, without training full scale models.

## A.3 In-Depth Evaluation Description

### A.3.1 Spoken Question Answering

**HeySquad**    HeySquad Wu et al. (2023b) is a spoken question answering (QA) dataset that aims to measure the QA ability of digital agents. It is based on the SQuAD dataset Rajpurkar et al. (2016) with 76K human-spoken and 97K machine-generated questions, and the corresponding answers. We evaluate the models on the open-source validation set with around 4K QA pairs.

**Spoken Dialect Question Answering (SDQA)**    SDQA Faisal et al. (2021) assesses the robustness of Spoken Language Understanding to global phonological variation in English. The dataset is made up of the same 1000 questions spoken and recorded by speakers in 10 accent regions where English is frequently spoken. We evaluate on the 494 of these questions which contain ground truth answers.

### A.3.2 Speech Classification

**Emotion Recognition    Interactive Emotional Dyadic Motion Capture (IEMOCAP)** IEMO-CAP Busso et al. (2008) is a dataset of ∼12 hours of videos, audio, motion capture, and transcripts of actors performing both improvised and scripted scenes. The seven professional and three student actors perform emotionally expressive scenes. Each conversation turn in each scene was labeled by six evaluators as demonstrating "happiness," "sadness," "anger," "surprise," "fear," "disgust," "frustration," "excitement," "neutral state," or "other." We follow Yang et al. (2024) and remove unbalanced class labels, resulting in 1241 audio utterances in the fifth fold used by Tang et al. (2023).

**Multimodal EmotionLines Dataset (MELD)** MELD Poria et al. (2019) contains 13,708 utterances labeled by emotion and collected from the sitcom *Friends*. MELD builds on EmotionLines Hsu et al. (2018); however, the authors of MELD ask annotators to watch the videos instead of simply reading the transcripts to produce labels. Three graduate student annotators labeled all utterances for emotions: "anger," "disgust," "fear," "joy," "neutral," "sadness," and "surprise," as well as for sentiments "positive," "negative," "neutral." We evaluate on the test set of 2608 utterances.

**Communicative Intent Recognition    Multimodal Sarcasm Dataset (MUSTARD)** MUS-TARD Castro et al. (2019) is a collection of 690 clips from the TV shows Friends, The Golden Girls, The Big Bang Theory, and Sarcasmaholics Anonymous, labeled as sarcastic or non-sarcastic by three annotators. The clips were collected primarily from YouTube using keywords like *Chandler sarcasm, Friends sarcasm, etc.* and sampled from MELD Poria et al. (2019). The final dataset was filtered to have an even number of labels of sarcastic and non-sarcastic clips. We evaluate on all 690 clips to test the models' capability in understanding intended sarcasm.

**URFunny** URFunny (Hasan et al., 2019) is a multimodal humor recognition benchmark constructed from 90.23 hours of TED talk recordings, spanning 1741 speakers and 417 topics. TED produces transcripts for the talks, which contain "[laughter]" markers that show when the audience laughs. The authors sampled the context and punchline before laughter markers for 8257 positive examples and random parts of the transcript without laughter markers for 8257 negative examples. URFun-nyV2 filters out noise and reduces overlap in examples. We evaluate 7614 examples from the train split of URFunnyV2 to evaluate the models' ability to understand speakers' humorous intents.

### A.3.3 Speech Translation

**CoVoST 2**    CoVoST 2 (Wang et al., 2020) is a speech-to-text translation benchmark to and from English. The speech inputs are sourced from the CommonVoice and professional translators are hired to translate the recording into a target language. The test dataset is large, made up of 15,500 examples translated from English to each target language. We evaluate on 7 target languages selected for their typological diversity in prior work (Clark et al., 2020).

## A.4 LANGUAGE ID OUTPUTS FOR ALL MODELS

Table 2: Percentage of outputs for which Language ID matches the target language

| Language | DiVA | KL Only | Token Alignment | Qwen | Qwen 2 | SALMONN |
|---|---|---|---|---|---|---|
| Arabic | 84% | 2% | 0% | 95% | 90% | 19% |
| German | 90% | 1% | 0% | 99% | 98% | 77% |
| Indonesian | 85% | 1% | 0% | 97% | 97% | 77% |
| Japanese | 28% | 2% | 0% | 100% | 99% | 67% |
| Tamil | 96% | 1% | 0% | 60% | 79% | 8% |
| Turkish | 74% | 1% | 0% | 93% | 92% | 28% |
| Mandarin | 60% | 2% | 0% | 91% | 83% | 93% |

## A.5 PROLIFIC USER DEMOGRAPHICS

Table 3: Aggregate Metrics for Age, Gender Identity, and High-Level Ethnicity Information from our User Study. Our participants cover a wide range of ages, are gender balanced, and have a similar distribution of ethnicities as reported in the United States Census.

| Age | | Gender Identity | | Ethnicity (Simplified) | |
|---|---|---|---|---|---|
| Median Age | 34 | Man | 50.9% | White | 59.2% |
| Max Age | 69 | Woman | 49.1% | Black | 16.3% |
| Minimum Age | 19 | Other | 0% | Asian | 12.2% |
| | | | | Mixed | 4.1% |
| | | | | Other | 8.2% |

