# OpenReview forum: "Distilling an End-to-End Voice Assistant Without Instruction Training Data"
_ICLR.cc/2025/Conference — Submitted to ICLR 2025_

### Official Review · Reviewer_WQdk · 2024-10-26

**Soundness:** 2
**Presentation:** 3
**Contribution:** 2
**Rating:** 3
**Confidence:** 4

**Summary:**

This paper proposes the Distilled Voice Assistant (DiVA), a Speech Large Language Model (LLM) trained on ASR data alone to enhance instruction-following abilities through self-supervised distillation. The authors align DiVA’s responses with those of a text-only LLM to enable cross-modal transfer of instruction adherence. However, the approach does not present substantive methodological innovation, as it closely mirrors existing distillation and instruction-following techniques. Furthermore, the work critically lacks comparisons to similar method in this domain, failing to substantiate its claimed improvements over key baseline methods.

**Strengths:**

- The use of ASR data alone to improve instruction-following behavior could offer cost benefits by reducing dependence on annotated instruction data.
- DiVA’s evaluation across diverse benchmarks provides a quantitative assessment of the model’s instruction-following performance.

**Weaknesses:**

- Lack of Novelty: The paper’s distillation approach lacks originality and does not go beyond established self-supervised and cross-modal transfer methods. Its reliance on prior distillation techniques without meaningful innovation limits the work's impact.
- Missing Comparisons to Key Similar Work: The paper does not compare DiVA to highly relevant models, such as those using cross-entropy on target sequences or alternative distillation methods, failing to clarify any advantages over standard methods in the field [1, 2, 3].
- Insufficient Literature Survey: The paper omits numerous relevant works, resulting in a limited survey that neglects essential context for DiVA’s approach within the broader Speech LLM and instruction-following literature.
- Paralinguistic Claims: Assertions about capturing paralinguistic features (e.g., sarcasm, emotion) are questionable given that DiVA’s speech embeddings are mapped to text-like embeddings, likely relying more on text semantics than true paralinguistic cues.



1. Fathullah, Yassir, et al. "AudioChatLlama: Towards General-Purpose Speech Abilities for LLMs." Proceedings of the 2024 Conference of the North American Chapter of the Association for Computational Linguistics: Human Language Technologies (Volume 1: Long Papers). 2024.
2.  Wang, Chen, et al. "Blsp: Bootstrapping language-speech pre-training via behavior alignment of continuation writing." arXiv preprint arXiv:2309.00916 (2023).
3. Wang, Chen, et al. "BLSP-KD: Bootstrapping Language-Speech Pre-training via Knowledge Distillation." arXiv preprint arXiv:2405.19041 (2024).

**Questions:**

- The authors should clarify the differences between this model and an ASR-LLM cascade, as mapping speech to text-like tokens questions both novelty and latency gains.
- Single-token predictions seem insufficient to capture temporal dynamics, potentially reducing DiVA’s effectiveness for tasks that require detailed temporal context.

---

> ### Author Response · Authors · 2024-11-17
> **Differences between DiVA and an ASR-LLM Cascade (Question 1)**
>
> We'd love to clarify the differences in DiVA in terms of modeling differences, latency, and novel functionalities compared to a cascaded system! These are grounded in some elements of our most recent response to Reviewer uKgu as well as an expansion on the reasoning presented on line 51-53 and 65-67 of our introduction.
>
> - **In Terms of Modeling**: Only the final N tokens are directly aligned to text-like tokens, which is the vast minority of the total tokens produced by DiVA (fewer than 5% of tokens for the average CommonVoice input, fewer than 10% of tokens for the longest input in CommonVoice).  Despite this small number, our ablations show that having some text-like tokens is key to enabling the model to follow text instructions included in the system prompt. While some of the tokens are directly text aligned for this purpose, the large number of non-text aligned tokens do not have a clear correspondence with a cascaded system.
>
> -  **In Terms of Latency**: An ASR-LLM Cascade with Whisper requires $N$ forward passes from the Decoder to produce a full transcript, where $N$ is the length of the true transcript. These sequential forward passes in the decoder are the dominant factor in the latency of autoregressive ASR systems such as Whisper [1, 2]. In a cascaded system, the first token generation is blocked on the transcription completing causing variability and slowdowns in the [time-to-first-token metric](https://docs.nvidia.com/nim/benchmarking/llm/latest/metrics.html#time-to-first-token-ttft).  On the other hand, our method requires only a single forward pass from the decoder, allowing LLM response generation to begin in constant time.
>
> - **In Terms of Capabilities**: Any ASR-LLM Cascade requires non-differentiable decoding steps for use. As such, a Cascaded system cannot be finetuned in an end-to-end fashion. This eliminates the potential use of such a system with increasingly utilized post-training methods dependent on gradients such as RLHF or DPO. Being differentiable all the way to the input spectrogram is a core capability which DiVA provides that a cascaded system does not, especially as differentiability is key to enabling a model to be post-trained and adapted to other tasks in line with the goals of the "foundation or frontier models" track of ICLR.
>
> [1] Gandhi, Sanchit, Patrick von Platen, and Alexander M. Rush. "Distil-whisper: Robust knowledge distillation via large-scale pseudo labelling." arXiv preprint arXiv:2311.00430 (2023).
>
> [2] OpenAI. "Whisper-V3-Turbo Model Release." GitHub, October 1st, 2024, https://github.com/openai/whisper/discussions/2363

---

> ### Author Response · Authors · 2024-11-19
> **On Temporal Context (Question 2)**
>
> We would love to get more information on this concern to respond more substantively. A priori, it is not clear to us how using a single token for output distillation would impact the ability for the model to capture temporal context, since gradients from this output token are backpropogated to the entire signal from the input audio.
>
> Temporal changes have multiple pathways to affect this output token:
> - The Whisper Encoder itself has a self-attention mechanism between the audio-tokens it produces.
> - Our Q-Former has bidirectional cross-attention from it's output tokens to all input tokens, allowing each token to accumulate information from any time in the input sequence and even across time.
> - Our Q-former has causal attention from the later tokens to all previous tokens, allowing tokens to accumulate information from earlier tokens.
>
> The ability of a Q-Former to aggregate information from non-sequential time steps is among it's benefits v.s. static pooling methods. As such, we would appreciate some expansion on this critique.
>
> Most concretely, could you provide an example of a dataset containing a task you are concerned about? If possible within the time constraint of the discussion period, we would love the opportunity to address your concern empirically with an evaluation on such a task.

---

> ### Author Response · Authors · 2024-11-27
> **Request for Response given the extended discussion period**
>
> Thanks again for your review! We have run a significant number of additional experiments in direct response to questions and concerns you posed in your review and we would really appreciate hearing from you in the discussion period!
>
> - Do the additional results address your concerns on missing baselines? Reviewer ukGu notes in their response that "It is now clear that the aligning method proposed in DiVA [does] have its advantage" over BLSP, BLSP-KD, and AudioChatLlama, but we would love to hear if that's the case for you as well since this was their most significant concern in your weaknesses. If not, we would love to understand what is still missing in your mind?
>
> - How could we address your concern on the temporal dynamics? We hope our explanation has made it clear how the model has the capability to incorporate temporal dynamics, but we would of course prefer to address your concern empirically! However, we are waiting to hear back from you on an example of a task you think DiVA would be unable to handle! We would really appreciate a response soon so that we may have time to run experiments given the new extended discussion timeline.

---

### Official Review · Reviewer_uKgu · 2024-10-27

**Soundness:** 3
**Presentation:** 2
**Contribution:** 1
**Rating:** 5
**Confidence:** 4

**Summary:**

This paper proposes a novel approach for training Speech LLMs in the absence of explicit instruction-following data. The authors introduce a method to transfer the instruction-following and conversational capabilities of text-based LLMs to speech-based models by leveraging only ASR data. Specifically, the proposed approach aligns input tokens using a cross-modal token alignment loss and output representations via embedding distillation loss, effectively bridging the modality gap between speech and text. Experimental results demonstrate that this method generalizes well to downstream tasks, including spoken QA and speech translation.

**Strengths:**

1.  The cross-modal alignment approach effectively retains general instruction-following abilities, addressing the common issue of forgetting in supervised fine-tuning (SFT).
2.  The Distilled Voice Assistant (DiVA) surpasses previous SOTA models, such as Qwen2-Audio, with significantly lower resource requirements.
3.  A qualitative user study shows DiVA’s strong alignment with human preferences in conversational quality.
4.  Ablation analysis confirms the distinct contributions of cross-modal token alignment and embedding distillation loss to DiVA’s performance.

**Weaknesses:**

1.  **Limited Novelty:** Although the authors suggest an "alternative paradigm" for aligning speech inputs to text-based LLMs, similar approaches involving distillation text responses for cross-modal alignment have already been explored in works like BLSP [1] and AudioChatLlama [2]. A more detailed discussion and comparison with these works would help clarify the unique contributions.
2.  **Potential Limitations of Q-Former as Modality Adapter:** DiVA employs a Q-Former as the modality adapter to convert Whisper outputs into tokens. However, recent research in Vision-Language Models suggests that Q-Former can introduce semantic deficiency and "redundant double abstraction," making it less effective than simpler alternatives like MLP with average pooling [3]. Furthermore, the fixed number of query tokens restricts the model’s ability to process speech of varying lengths, potentially limiting its adaptability.
3.  **Unclear Basis for Performance Gains:** The claim that DiVA outperforms SOTA models like Qwen2-Audio may be misleading, given that DiVA builds on Llama-3—a stronger backbone than Qwen2-Audio’s Qwen-7B. This discrepancy makes it difficult to attribute performance gains solely to the proposed training method. Including the text-only performance of the backbone models (Llama-3 / Qwen-7B) or training DiVA on the same backbone as Qwen2-Audio could clarify the source of the observed improvements.



[1] Wang C, Liao M, Huang Z, et al. Blsp: Bootstrapping language-speech pre-training via behavior alignment of continuation writing[J]. arXiv preprint arXiv:2309.00916, 2023.

[2] Fathullah Y, Wu C, Lakomkin E, et al. AudioChatLlama: Towards General-Purpose Speech Abilities for LLMs[C]//Proceedings of the 2024 Conference of the North American Chapter of the Association for Computational Linguistics: Human Language Technologies (Volume 1: Long Papers). 2024: 5522-5532.

[3] Yao L, Li L, Ren S, et al. DeCo: Decoupling Token Compression from Semantic Abstraction in Multimodal Large Language Models[J]. arXiv preprint arXiv:2405.20985, 2024.

**Questions:**

Questions:
1. In Section 3.1, the authors state that they use the decoder weights of Whisper to initialize the Q-Former’s _cross attention_. However, in Section 3.2.1, it’s mentioned that the speech tokens are processed with _causal attention_ in Whisper's decoder. This seems contradictory and is confusing. Could the author clarify the architecture of the Q-Former adapter?
2. In Section 3.2.1, the cross-modal token alignment loss is applied only to the last $N$ tokens. However, this choice is unclear since Q-Former’s output tokens lack causal relationships. Additionally, the number of speech tokens $Q$ is defined as a hyperparameter, so the assumption that $Q>N$ may not hold for all inputs during inference. Could the author provide further justification for these design choices and proofs for the assumption about token counts?
3. In Section 3.2.1, the author claim that the additional $Q-N$ token provide information bandwidth for other information. However, I see no evidence to support this claim. Is there any statistic on the number of tokens that did not undergo alignment during training? If so, how does dropping these tokens affect the model's performance?
4. In Section 3.2.2, the authors claim that the proposed $L_2$ loss can be computed more efficiently than KL divergence. Could additional evidence on training costs (e.g., computation time or resource usage) be provided to support this claim?
5. The ablation study indicates that the token alignment loss aids the model in adhering to text instructions. However, this difficulty in following text instructions might stem from DiVA being trained solely on speech inputs without accompanying text instructions. Could incorporating text instructions before speech tokens during training improve performance? Furthermore, if this modification were implemented, would the token alignment loss still be necessary, or could it be reduced or omitted?

Typos and minor mistakes:
1. In Table 1, the base LLM for Qwen 2 Audio Instruct is listed as Qwen2 Instruct. However, according to the technical report, the correct base model should be Qwen-7B.
2. In Equation (1), the subscripts in the summation notation should start from 1.
3. In Section 3.2 line 247, "L2" should be $L_2$

[1] Chu Y, Xu J, Yang Q, et al. Qwen2-audio technical report[J]. arXiv preprint arXiv:2407.10759, 2024.

---

> ### Author Response · Authors · 2024-11-16
> **On the cost of KL Divergence (Question 4)**
>
> We'd like to expand the statement that $L_2$ loss on output embeddings is computationally cheaper than KL divergence over the vocabulary! In practice, the difference comes down to which dimension of the model the operations are dependent on. The two dimension of import are the embedding dimension $d$ and the vocabulary dimension $V$. As we note on line 248, in Llama 8B and other similar models $d$ is roughly an order of magnitude smaller than $V$ ($d$=8192, $V$=128k for Llama 3 8B).
>
> Up until the output, the cost of each loss is identical. Since the $L_2$ loss uses the output directly at this phase, it incurs an O(d) cost for the subtractions and multiplications required to compute the $L_2$. On the other hand, the KL Divergence O($V\cdot d$) cost since you must multiply the output embedding with the vocabulary embedding matrix and then take the difference between the two $V$ dimensional prediction distributions.
>
> Using the $L_2$ loss removes the dependency on the vocabulary size, which is the larger part of the cost, without harming the quality of fit as we show in Apendix A.2.

---

> ### Author Response · Authors · 2024-11-16
> **On the Whisper Decoder Initialized Q-Former Architecture (Question 1)**
>
> We'd love to resolve the confusion that led to an apparent contradiction in the Q-Former! The architecture of our Q-Former is identical to the Decoder from Whisper, which makes no modifications to the original transformer from "Attention is All You Need".
>
> The Whisper Decoder contains two distinct attention layers:
> - A causal self-attention layer, which we refer to in section 3.2.1. In this layer, K, Q, and V are all projections of the Decoder inputs, with causal masking ensuring tokens only attend to previous positions.
> - A non-causal cross-attention layer, mentioned in section 3.1, where Q comes from the Decoder inputs while K and V are projections of embeddings from the Whisper Encoder.
>
> We understand how references to these separate attention mechanisms could seem contradictory without explicitly clarifying the Whisper Decoder architecture. We will update Figure 1 to illustrate both attention mechanisms and their information flow to make this clearer.

---

> ### Author Response · Authors · 2024-11-17
> **On Our $Q>N$ Assumption and Generalization To Lengths Longer than 30 Seconds (Part of Weakness 2 and Question 2)**
>
> The Whisper encoder itself is "trained on 30-second audio chunks and cannot consume longer audio inputs at once" [1]. As such, all Speech LLMs which utilize this frozen encoder – which includes Qwen 1 & 2, SALMONN, BLSP, and BLSP-KD – inherit this constraint. We keep the maximum of 448 tokens as output used in the Whisper paper for transcription, which is based on the heuristic assumption that speech is unlikely to exceed 14 tokens per second. Human speech has been shown to transmit roughly "39 bits per second" of information across languages [2], so this assumption is theoretically justified as 14 tokens per second allows for up to 112 bits per second even if we make the conservative assumption that each embedding only encodes a single byte of information.
>
> We confirmed that this also empirically reasonable by computing the token statistics from the entire CommonVoice eval dataset, which covers 16.4k utterances:
>
>
> | Metric         | \# of Tokens |
> | -------------- | ------------ |
> | Mean           | 15.218018    |
> | STD            | 4.082178     |
> | Max            | 39           |
> | 75% Percentile | 18           |
> | 50% Percentile | 15           |
> | 25% Percentile | 12           |
> | Min            | 5            |
>
> As this shows, Q > N is a relatively weak assumption for Q=448, with the longest sequence in CommonVoice being 39 tokens. Given that Q is significantly larger than N, the vast majority of the input tokens are *not* directly aligned with text tokens during the training process. Importantly, DiVA requires only one forward pass for all 448 tokens while a Cascaded Model requires $N$ auto-regressive forward passes from the Decoder based on the true transcript length.
>
> At inference time, for recordings of greater than 30 seconds or multi-turn conversations, we follow the paradigm used by Whisper itself for transcription and separately encode chunks of audio. The tokens from each of these chunks are fed together as input to the backbone LLM for audio longer than 30 seconds. This chunking approach is necessary even for methods which use an alternative adapter, such as an MLP, between Whisper and an LLM since the 30 second cap comes directly from the Whisper encoder not from the Q-Former.
>
> [1] Radford, Alec, et al. "Robust speech recognition via large-scale weak supervision." International conference on machine learning. PMLR, 2023.
>
> [2] Coupé, Christophe, et al. "Different languages, similar encoding efficiency: Comparable information rates across the human communicative niche." Science advances 5.9 (2019): eaaw2594.

---

> ### Author Response · Authors · 2024-11-27
> **Any further questions given the extended discussion period?**
>
> Thank you for your review and for responding to our additional baseline experiments to acknowledge that the advantages of DiVA are now clear!
>
> We hope the above responses, including additional requested metrics, to your other extended questions have clarified the points! If they haven't, we would love to hear how they miss the mark and if there is anything we can still clarify. If they have clarified these points, in addition to the concerns on the baselines, we would love to know what open concerns have kept your score unchanged from the original review and if there are any concrete ways we can address them.

---

> > ### Comment · Reviewer_uKgu · 2024-11-30
> >
> > Thank you for your reply.
> >
> > I think the additional experiments help address most of my concern. While the proposed approach may exhibits better performance compared to previous works such as BLSP, relying on self-knowledge distillation to reduce the forgetting of text-LLM (the core contribution claimed in the paper) have already been explored by these works. Changing token-level KD to representation-level has also been widely discussed. Therefore, I found the novelty of the proposed paper limited. I would like to maintain my current score.

---

### Official Review · Reviewer_RRHC · 2024-10-27

**Soundness:** 3
**Presentation:** 4
**Contribution:** 4
**Rating:** 6
**Confidence:** 4

**Summary:**

This paper proposes a method named DiVA for training a speechLLM without using any instruction data in the speech modality. To perform this, an LLM is adapted to enable speech input and output by first combining it with a Whisper based speech encoder, and then using two Novel loss term to train the new modules. The two losses are a cross-model alignment loss which aims to alight the speech and text modalities, and a (simplified) KL divergence loss on the distribution of text and speech outputs. The authors evaluate the paper on 3 groups of tasks, which are spoken question answering, speech classification and speech to speech translation. In all 3 task groups the method reaches state of the art results and surpassed the baseline methods.

Overall, this paper shows a significant contribution and I recommend to accept it.

**Strengths:**

1. The proposed method is novel and easy to reproduce.
2. Evaluations show strong results on a wide variety of tasks.
3. The paper is clearly written.
4. Authors use publicly available datasets and fine-tuned models.

**Weaknesses:**

1. It is unclear how speech (waveform) is generated from the model output.
2. The paper is missing some direct evaluation of the speech quality, such as MOS experiments.
3. For the speech to speech translation task, the paper is lacking an evaluation of the speaker similarity and translation quality (ASR-BLEU) metrics, this may better inform the reader about the overall quality of the translations.
4. While the paper claims to have created a voice assistant, there is no evaluation or clarification what a voice assistant means. It isn't clear that good performance on the tasks stated in the papers results in a voice assistant. This should be clarified or rephrase.

**Questions:**

1. How is speech generated from the model output?
2. How is the quality of the output speech compared with the ground truth speech?
3. it is stated that without the KL-divergence loss the speech output is incoherent, how was this measured?
4. Are there any audio samples from the model?
5. In what voices is the model able to generate speech?

---

### Official Review · Reviewer_Rub3 · 2024-11-06

**Soundness:** 3
**Presentation:** 3
**Contribution:** 2
**Rating:** 5
**Confidence:** 4

**Summary:**

This paper proposes DiVA, a voice assistant model that is able to follow both spoken and written instructions. It is trained via a dual distillation and alignment loss and shows relatively strong results on various benchmarks including head to head comparison with Qwen 2 Audio. The authors propose a "q-former" style injection of audio into the model, which is initialized from a Whiper model, and a text model initialized with Llama weights and left frozen which consumes the audio input. This model is then trained for relatively small amount of steps on CommonVoice data to learn input alignment as well as output alignment (as compared to text labels).

**Strengths:**

- Interesting approach which takes several pre-trained models (Llama, Whisper), connects them via a q-former based adapter and trains a small portion of the combined model to respond to audio inputs.
- Scores relatively well on various benchmarks compared to some well known models.
- The main contribution is showing that it is sufficient to teach an existing instruction-tuned model to understand audio through relatively light weight techniques which then allows the model to follow instructions via audio queries.

**Weaknesses:**

There is not a lot of novelty in this method besides choosing which data to train on. Q-former or in general attention based pooling (e.g. as in perceiver) is well known; L2 loss as a replacement for KLD has also been around for awhile (e.g. in Soundnet: Learning sound representations from unlabeled video.). Putting some of these pieces together to allow Llama models to process audio input has also been explored, for example in Nvidia's SpeechLLM or SALMONN (which is cited in this paper and apparently does not use Llama). In my opinion demonstrating that full SFT is not required to obtain an audio understanding model is interesting but not sufficient.

**Questions:**

- Have you compared your approach with an ASR -> LLM based pipeline? One would imagine it would score poorly on the emotion recognition tasks, but probably very high on others.
- Have you considered diving deeper into what kind of data is requires to achieve specific results? For example, you could ablate the amount of commonvoice data required to achieve specific results; explore other datasets and/or languages and measure their impact on downstream tasks. This would provide a valuable contribution to the community.

---

> ### Author Response · Authors · 2024-11-19
> **On The Lack of Novelty in The Use of $L_2$ Instead of KLD (Weakness Item 2)**
>
> Thank you for drawing this interesting connection to SoundNet's experiments using $L_2$ loss. While both approaches use $L_2$ distance, they apply them at different parts of their respective models which leads to different implications:
>
> - SoundNet "experimented with using $L_2$ loss on the target outputs instead of KL loss", though they found KL divergence worked better in their architecture [1]. In the notation of our paper, this would be $||P_t - P_s||_2$
> - Our work does not use the $L_2$ loss as a drop-in replacement for the KL Divergence, but applies it instead to the hidden states $||h_t - h_s||_2$. This removes the need to compute $P_s = \sigma(O_sh_s)$ entirely. This shift to hidden states is only justified since the output vocabulary embeddings $O$ are shared and frozen between teacher and student in our work - a design very different from SoundNet's two separate networks.
>
> Our usage of the $L_2$ loss leads to one key advantage and requires one key additional proof when compared with the SoundNet work:
>
> - **Performance Improvement**: Operating on hidden states (dimension $d$) rather than full probability distributions (vocabulary size $V$) reduces computation from $O(V \cdot d)$ to $O(d)$. With $d=8196$ and $V=128k$ in our implementation, this is an order of magnitude speedup in loss computation.
> - **Theoretical Justification**: Since our approach removes several steps of computation entirely, it is not as straightforward to see that it nonetheless leads to a global minima of the KL Divergence. In Lemma 1, we show the conditions required for this adaptation to be theoretically justified. We further confirm the theory is practically predictive with optimization experiments in Appendix A.2.
>
> Notably, this adjustment resolves a performance issue mentioned explicitly in prior work on context distillation: "However,
> this leads to another issue: the vocabulary space of language models is often on the order of 50-100k, and the full soft labels consume a lot of memory" [2]. By comparison, the form of $L_2$ loss used in SoundNet would not resolve this issue since the full soft labels are still required.
>
> [1] Aytar, Yusuf, Carl Vondrick, and Antonio Torralba. "Soundnet: Learning sound representations from unlabeled video." Advances in neural information processing systems 29 (2016).
>
> [2] Snell, Charlie, Dan Klein, and Ruiqi Zhong. "Learning by distilling context." arXiv preprint arXiv:2209.15189 (2022).

---

> ### Author Response · Authors · 2024-11-27
> **Request for response given the extended discussion period**
>
> Thanks again for your review! We have run a set of new ablations in response to your concerns and we would really appreciate hearing from you during the discussion period to understand whether these address your concerns and, if not, how they could be addressed
>
> - Do the additional ablations on the connector method clarify the contribution of our initialization method described on lines 176-181? We have now shown that this initialization, while simple, meaningfully improves performance! As you note, other works on Speech LLMs use Q-Formers, so this initialization approach would likely improve even non-distillation based methods.
>
> - Is the distinction between our usage of the $L_2$ loss from SoundNet's clearer? Notably, as we describe above, while both methods use the $L_2$ loss, our method is distinct and offers performance improvements that SoundNet's method does not by exploiting specific properties of encoder training with a frozen LLM that do not exist in the SoundNet architecture.
>
> - Given the now extended set of ablations, is our improvement above other models which add speech encoders to LLMs clearer? Our work goes beyond showing that training with existing audio SFT data is "not required" by showing that training with existing audio SFT data is **worse** than our method using only ASR data. In our original, we show this compared to SALMONN, Qwen Audio, and Qwen Audio 2 which are the state-of-the-art SFT Speech LLMs and we have now shown this compared to several other requested baselines.

---

> ### Comment · Reviewer_Rub3 · 2024-11-28
>
> Thanks for the response and the several new experimental results.
>
> > Do the additional ablations on the connector method clarify the contribution of our initialization method described on lines 176-181?
>
> While additional ablations are helpful, I think the fact that you initialize q-former with a pre-trained model weights is not really in and of itself a novel technique worthy of a paper. However it is definitely a noteworthy and useful method as part of a greater work.
>
> > Is the distinction between our usage of the L2 loss from SoundNet's clearer?
>
> While your proposal is somewhat different, there are many papers (e.g. [1], [2], just from cursory browsing) that have some form of feature distillation or some variation of an L2-type loss. It is not a big leap to go from soundnet or other works to a feature-regression like loss that you propose.
>
> [1] https://arxiv.org/pdf/2203.10163
> [2] https://arxiv.org/pdf/1811.03233
>
> > Given the now extended set of ablations, is our improvement above other models which add speech encoders to LLMs clearer?
>
> I appreciate the new ablations and it does indeed show that this proposal appears to outperform some contemporary works. However these comparisons are not complete as they are missing comparison to other adapters on top of Llama3. As your ablations show, using Llama3 as a backbone significantly outperforms other models even in a cascaded fashion.
>
> As you have added additional results that show strong performance (which I recommend adding to the paper as well) , I will slightly raise my score. However I still believe that the combination of techniques in this paper, outside of data selection, is of limited novelty for practitioners in the field.

---

> > ### Author Response · Authors · 2024-12-04
> > **On Additional Feature Distillation Citations**
> >
> > Thank you for the additional citations on feature distillation in the image classification domain. We'd like to note a key difference in these works. These works study model compression in which "the number of neurons might be different" [1] since "the student and teacher have different types of neural network architectures [or] the student has a smaller model width" [2]. As such, the first work "use[s] a connector function that consists of simple operations" [1] and the second uses "a linear transformation r on the outputs of gs(x) to match the teacher’s dimension" [2]. In this setting, Lemma 1 in our work is not applicable due to the mismatch in dimensions.
> >
> > We do agree that the analysis from the second work which concludes that "minimizing the difference between the student’s and teacher’s intermediate representations reduces the KL divergence" [2] and their experiments on incremental learning is notable in regards to our work. We understand your feeling that this may make this contribution not "a big leap". However, we hope that this "somewhat different" formulation as well as the thorough experiments showing it's effectiveness in a different setting of adding a modality to LLMs is useful to practitioners.
> >
> > As you note more broadly, we especially hope this is useful as a part of the larger set of methods and decisions that make DiVA effective such as the model initialization, the combination of both input and output distillation, and the data selection which you note.
> >
> > [1] https://arxiv.org/pdf/2203.10163
> > [2] https://arxiv.org/pdf/1811.03233

---

### Author Response · Authors · 2024-11-15
**On Request for Additional Comparisons to Other Models**

We thank the reviewers for their comments! We are working hard to provide substantive responses backed with experimental results to address reviewer concerns wherever possible. We will continue to provide concrete responses to these throughout the discussion period! The first of these empirical results is in regards to the reviewer request for further baseline comparisons to other recent Speech LLMS. Where possible we have run additional evaluations to directly address this feedback!

First, we focus on specific points surrounding particular models that the reviewers have requested comparisons to. These models are BLSP, BLSP-KD, and AudioChatLlama:

1. We have run our evaluations on BLSP, which performs hard-distillation on the outputs using cross-entropy loss, showing that DiVA improves performance across all metrics (see Table below) compared with this model.
| Task        | DiVA   | BLSP   |
| ----------- | ------ | ------ |
| SDQA-Mean   | **53.58%** | 44.03% |
| HeySquad    | **55.17%** | 46.78% |
| IEMOCAPS    | **50.60**  | 42.88  |
| MELD        | **41.34**  | 40.12  |
| UrFunny     | **50.24%** | 49.92% |
| MUSTARD     | **52.61%** | 48.41% |
| COVOST-Mean | **13.80**  | 5.05   |

2. Neither BLSP-KD (published on Arxiv on May 29, 2024) and AudioChatLlama (published in NAACL on June 15, 2024) release their model weights or the code used to produce these models. We have emailed the authors of these works to see if we can get access to perform these comparisons. The BLSP-KD author unfortunately said that they cannot release their model as it was produced during a company internship. We are still waiting on a reply front the AudioChat Llama author to reply and will respond further when they do.
\
Without code or model weights it is difficult to compare directly to these recently released works, highlighting the importance of our entirely open-source pipeline for this work. However, we have done our best to provide empirical comparisons to these works based on available information in the next two points.

3. Our "Distillation Only" ablation closely mirrors AudioChatLlama's described methodology, which performs distillation on outputs only, revealing important limitations in generalization. Specifically, while this approach proves effective for Spoken Question-Answering (as both our works show), our broader evaluation reveals that it fails to generalize to tasks requiring text instruction following. This highlights the importance of our usage of both input and output distribution alignment.

4. Without access to the particular model, we instead compare the results from our model on the same COVOST languages they evaluate on in BLSP-KD to the results they report in their paper. We show this comparison below, where DiVA outperforms BLSP-KD on 5/7 languages used in the BLSP-KD paper.

| Model              | en-ca | en-de | en-id | en-ja | en-sl | en-sv | en-zh |
| ------------------ | ----- | ----- | ----- | ----- | ----- | ----- | ----- |
| BLSP-KD (Reported) | 23.5  | 24.4  | 13.4  | **21.3**  | 10.7  | 24.5  | **41.3**  |
| DiVA               | **29.07** | **27.6**  | **22.8**  | 6.2   | **11.2**  | **24.7**  | 12.2  |

As noted in our paper, Llama (and therefore DiVA) produces romanized, rather than native script, responses in Chinese and Japanese which leads to the reduced performance in those languages compared to BLSP-KD.

There are always many models which can be included as baselines, especially in quickly growing research areas. We feel our selection of SALMONN, Qwen Audio, and Qwen 2 Audio as primary baselines instead is supported by:
1. Their state-of-the-art performance on chat-style benchmarks and paralinguistic tasks as shown in the recent Qwen 2 technical report in which these are the top 3 performing models [1].
2. Open availability of model weights and inference code which makes evaluation of them possible without re-implementing and retraining the models.
3. Their widespread usage overall as the most commonly used reference point and/or foundation models for other Speech LLM works. This is also likely due to their open-access release which makes their usage possible in downstream research, similar to DiVA.

[1] Chu, Y., Xu, J., Yang, Q., Wei, H., Wei, X., Guo, Z., ... & Zhou, J. (2024). Qwen2-audio technical report. arXiv preprint arXiv:2407.10759.

---

> ### Author Response · Authors · 2024-11-15
> **On Request for Additional Comparisons to Other Models (Cont)**
>
> Our method varies from BLSP, BLSP-KD, and AudioChatLlama in our usage of L2 losses rather than either hard-distillation through cross-entropy or through direct optimization of KL divergence which we show is computationally inefficient and unneccesary. Furthermore, our extensive evaluation offers new insights beyond what BLSP, BLSP-KD, and AudioChatLlama show.
>
> 1. Our work is the first direct comparison between distillation and SFT approaches. BLSP, BLSP-KD, and AudioChatLlama compare to their own ablation based models but not to the SFT models that are currently state-of-the-art for end-to-end models.
> 2. Evaluation of a model trained using cross-modal distillation on paralinguistic tasks. BLSP-KD and AudioChatLlama evaluate on QA and Speech Translation tasks, but not on tasks that benefit from paralinguistic information. Directly, we have now shown that BLSP underperforms DiVA on paralinguistic tasks.
>
> 3. Novel human judgment evaluation showing distillation outperforms SOTA SFT models. We recruit annotators to directly compare the resulting models in their actual desired use-cases, giving us a better understanding of the usability of our model, rather than performance on a static benchmark with a specific prompt format.

---

> ### Comment · Reviewer_uKgu · 2024-11-17
> **Question regarding the backbone model.**
>
> Thank you for your response. The provided results show that DiVA outperforms BLSP and BLSP-KD. However, the current comparison is mainly conducted at the system level, where the difference in backbone model capacity may also contribute to the performance gain. I suggest reproducing BLSP with Llama-3 backbone or implementing DiVA on Qwen-Instruct, which could provide a more fair comparison.

---

> ### Author Response · Authors · 2024-11-17
> **On the Differences in Backbone Models**
>
> We will attempt to train a DiVA version of Qwen if possible within the discussion period. This may not be feasible within this time constraint however as DiVA's training relies on free computing resources from the TPU Research Cloud and therefore our code is in Jax. Unfortunately, there is not yet a Jax implementation of the Qwen models available and this may take us some time to re-implement and debug.
>
> In the meantime, we can reasonably show that the improvements are not simply due to a better backbone model by comparing the performance of the text-only models as you suggested in your review! Here, we compare to Qwen (backbone for Qwen and Qwen 2 Audio), Llama 2 (backbone for BLSP), and Llama 3 (backbone for DiVA) on one of our Question Answering tasks Spoken Dialect QA.
>
> | SDQA-Set      | DiVA Improvement over BLSP (*E2E*) | DiVA Improvement over Qwen 2 Audio (*E2E*) | DiVA to Qwen Audio (*E2E*) | Llama 3 Improvement over Qwen (*Cascaded*) | Llama 3 Improvement over Llama 2 (*Cascaded*) |
> | ------------- | -------------------------- | ---------------------------------- | ------------------ | ---------------------------------------- | ------------------------------------------- |
> | Australia     | +9.44                      | +11.15                             | +10.11             | +5.16                                    | +3.29                                       |
> | New Zealand   | +10.56                     | +10.01                             | +10.67             | +5.21                                    | +3.55                                       |
> | Great Britain | +9.03                      | +9.96                              | +10.93             | +5.11                                    | +3.03                                       |
> | United States | +9.80                      | +10.39                             | +12.31             | +5.54                                    | +3.80                                       |
> | Ireland       | +9.21                      | +10.66                             | +11.17             | +5.12                                    | +2.96                                       |
> | South India   | +10.31                     | +11.12                             | +10.94             | +4.70                                    | +3.09                                       |
> | South Africa  | +9.02                      | +9.95                              | +8.67              | +5.10                                    | +3.09                                       |
> | Phillipines   | +9.47                      | +9.75                              | +10.40             | +4.88                                    | +3.05                                       |
> | Nigeria       | +10.07                     | +11.19                             | +10.59             | +3.81                                    | +1.72                                       |
> | North India   | +8.58                      | +10.19                             | +9.51              | +4.81                                    | +2.30                                       |
>
> As you say, Llama 3 is indeed a stronger backbone model, which confirms our choice of it as reasonable! However, if the performance gains were only due to a stronger backbone model we would expect the improvements in DiVA to be similar or smaller than the improvements in the text-only setting.
>
> Instead, DiVA improves performance over it's baseline end-to-end models by a larger margin than the difference between the corresponding backbone models used as text-only LLMs with the same ASR model. This is especially notable as DiVA trains with less data than BLSP (\~8x less), Qwen Audio (\~10x less), or Qwen 2 Audio (\~100x less).

---

> ### Comment · Reviewer_uKgu · 2024-11-18
> **Maybe adding the ablation results is more helpful?**
>
> Thank you for your response. It is now clear that the aligning method proposed in DiVA do have its advantage. I would suggest adding the results with ablation to this table. Maybe we can then see which part of the method contributes more.

---

### Author Response · Authors · 2024-11-18
**On Our Use and Adaptation of the Q-Former Architecture**

In the second of our empirical results in response to reviewer comments, we'd like to address the concerns on our use of the Q-Former and clarify how our work improves over other usages of Q-formers.

We agree with Reviewer Rub3 that the Q-Former is indeed well-established in this domain, as acknowledged in our paper (lines 168-170). We also acknowledge Reviewer uKgu's point about potential limitations of the Q-Former! We noted ourselves that attention-based pooling typically increases data requirements compared to simpler heuristic pooling methods (lines 173-174).

## Novel Contribution

Our contribution on the particular topic of the architecture of the adapter is simple, but empirically effective: initializing the Q-Former using the Whisper Decoder with static query tokens (discussed on lines 176-181). While this approach limits our architecture choices somewhat, we hypothesized that leveraging the pretraining of the Whisper Decoder would be more practically beneficial. To provide stronger evidence for this hypothesis, we conducted additional empirical experiments.

## Experimental Setup

We performed three comparative training runs:

1. Our proposed approach
2. Control 1: Q-Former with random initialization
3. Control 2: MLP with random initialization

All other training parameters remained constant across runs including training data order, other component initializations, and hyperparameters.

## Results and Analysis

### Performance Across Tasks

We see that much lower [training](https://ibb.co/hZC32ky) and [evaluation](https://ibb.co/sKDptSy) loss are achieved by the approach we take in our paper v.s. either of the controls. We have attached the training and evaluation loss curves in anonymous links here.

Beyond lower loss, our empirical results demonstrate significant improvements across most tasks. Most notably, the Q-Former initialization is critical to achieving non-zero translation performance:

| Task Type | Task Name | Q-Former w/ Whisper Init | Q-Former w/ Random Init | MLP w/ Random Init |
|-----------|-----------|-------------------------|------------------------|-------------------|
| Question Answering | SD-QA | **53.58%** | 45.94% | 25.73% |
| Question Answering | HeySQUAD | **55.17%** | 47.63% | 29.48% |
| Emotion Recognition | MELD | **41.34** | 39.88 | 8.74 |
| Emotion Recognition | IEMOCAPS | **50.60** | 44.08 | 19.85 |
| Sarcasm Detection | Mustard | 52.61% | 55.07% | **56.52%** |
| Humor Detection  | URFunny | 50.24% | 50.05% | 50.28% |
| Translation | CoVOST | **13.80** | 0 | 0 |

Qualitatively, the MLP connector seems to learn only to produce the first token correctly and then immediately generates EOS afterwards. This leads to reasonable classification performance, but poor performance in generative tasks. On the other hand, the randomly initialized Q-Former is able to generate coherent QA responses, but fails to adapt to varied text-based instructions leading to zero BLEU scores in translation.

### A Note on Length Generalization

This limitation applies equally to MLP with average pooling since it produces tokens at a fixed-ratio to the input tokens. The Whisper Encoder (also used in Qwen 1 & 2, SALMONN, BLSP, and BLSP-KD) produces mixed 1500 tokens for all input lengths and has a maximum input length of 30 seconds. As noted in our response to Reviewer uKgu on the 448 token limit, in practice, this limitation can be mitigated both for DiVA and other models that use the Whisper Encoder by encoding multiple chunks of 30 seconds for length generalization.

## Conclusion

We hope that these additional experimental results address the reviewer concerns by showing that:

1. The Whisper-initialized Q-Former generally improves over the suggested simpler alternative from Reviewer uKgu
2. Our contribution on this particular front is not the architecture itself, but instead a simple but previously unused initialization strategy to reduce the data demands of attention-based pooling. This adjustment is likely to benefit not just DiVA, but other approaches which use the Q-Former for SFT based training.

---

### Author Response · Authors · 2024-12-04
**Final Comment of Review**

We would like to thank the reviewers for their time!  We are glad to hear from the reviewers that there are limited concerns about technical soundness after the discussion period, despite remaining questions about novelty. The main open soundness concerns are from Reviewer WQdk, who unfortunately we did not have a chance to discuss further with after their initial review. We hope that similar to Reviewer uKgu "the additional experiments help address most of [their] concerns", given that they had shared concerns about the baselines compared to.

While we understand the reviewers concern that the novelty of our work is not a "big leap", we want to highlight the following practical merits of our work:
- The reviewers agree that the model is clearly empirically effective and outperforms similarly situated models. We feel the practical utility of a strong open-weights model which can be used for interpretability studies, further refinement, and self hosted application can be significant.
- Releasing the full code to reproduce such a model on the openly-accessible infrastructure we used to train DiVA meaningfully contributes to open and reproducible science and enables future works leveraging the underlying code for scalable training.
- The ablations studying the effects of our varied contributions help practitioners understand each separately, allowing them to utilize different useful components even if they do not use all of the methods we study.

As the discussion period is coming to an end, we want to close with a final expansion on these points.

## DiVA has Strong Empirical Results For Downstream Usage:
- While individual components of our work may be seen to lack novelty, DiVA demonstrates a practical advance in speech-LLM integration. Following our additional baseline experiments, all of the reviewers who responded agreed that our model's performance is clearly stronger than other end-to-end speech models.
- Unlike some recent works the reviewers mentioned, our work releases model weights which can be finetuned further for specific tasks, styles, or languages. We contacted the authors of these works and either did not hear back or were told the weights will not be released due to corporate policy. Given these works seem unlikely to be released for further usage or study, releasing an open model with state-of-the-art capabilities is especially likely to enable new work from the broader community.
- Our open-source implementation supports reproducibility, which RRHC highlighted as a strength: "Authors use publicly available datasets and fine-tuned models". By releasing training code which uses computing infrastructure which is freely available to academics, our work could enable more practitioners to enter and advance this research topic.

## Multiple Methods, One Effective Model
- While Reviewer Rub3 noted that "using q-former with pre-trained model weights is not really in and of itself a novel technique," they acknowledged it is "definitely a noteworthy and useful method as part of a greater work". Notably, this initialization method is one part of what leads to DiVA's success and complements our separate distillation components.
- Reviewer uKgu notes that "ablation analysis confirms the distinct contributions of cross-modal token alignment and embedding distillation". This provides more nuance to understand why DiVA is so empirically effective with regards to other distillation works.
- While Reviewer Rub3 notes that other works have explored $L_2$ or feature based losses, we feel these works are meaningfully different from ours. The most notable difference is that, specific to our case of adding a new modality to an otherwise frozen LLM, we show that feature distillation leads to an equivalent solution to KL Divergence distillation at reduced cost due to the shared components.

## DiVA offers Comprehensive Evaluation and Analysis
- As demonstrated in our response tables, we conducted thorough comparisons across architectures, losses, and to all baselines which were requested and available. Across these, we have demonstrated that DiVA is effective and often exceeds the performance of models trained on far more data. We are glad to see that there are limited soundness concerns about which tasks we selected given DiVA achieves "strong results on [this] wide variety of tasks" [RRHC].
- As uKgu noted, this includes "a qualitative user study [which] shows DiVA's strong alignment with human preferences in conversational quality". This pairwise preference study for Speech LLMs is a first of its kind and we release the open-source code to enable future such studies by others. We hope that the contribution of this first pairwise preference study of Speech LLMs clearly highlights the benefits of this approach, since as we note on line 485, our results provide early indications that many current "benchmarks may not correlate with practical usage."

---

### Meta-Review · Area_Chair_14Vc · 2024-12-18

**Metareview:**

The paper introduces DiVA (Distilled Voice Assistant), a model for training speech-based large language models (Speech LLMs) without requiring explicit instruction-following data in the speech modality. The key components of the approach are: 1) integrating a Whisper-based speech encoder with a text model initialized from Llama weights, allowing the model to process both speech and text inputs. The text model remains frozen while the combined model is trained on a relatively small amount of data, including CommonVoice. 2) training the model using dual distillation and alignment loss techniques, with cross-modal alignment loss to align speech and text modalities and embedding distillation loss to align output representations of speech and text. With these components, DIVA demonstrates competitive performance across several task groups, including spoken question answering, speech classification, and speech-to-speech translation, and surpassing baseline methods, such as Qwen2-Audio.

Strength of this paper

- Competitive performance: DiVA surpasses previous SOTA models, like Qwen2-Audio, with significantly lower resource requirements, making it a more resource-efficient solution. Evaluations show competitive results across various tasks and benchmarks, with DiVA performing well in instruction-following and cross-modal alignment. These experiment results support the model’s efficacy.
- A cost-effective methodology of using ASR data alone to improve instruction-following behavior and reducing reliance on annotated instruction data. The paper also demonstrates that it is sufficient to teach an existing instruction-tuned model to understand audio using relatively lightweight techniques, effectively enabling the model to follow instructions via audio queries.
- The paper is clearly written, and it uses publicly available datasets and fine-tuned models, which increases the paper's accessibility and reproducibility.


Weakness of this paper

- Limited Novelty: the method lacks novelty, as many components (e.g., attention-based pooling like Q-former, L2 loss as a replacement for KLD) are well-known in existing literature. Combining these pieces to process audio input with Llama models, though interesting, has been explored in previous works like Nvidia's SpeechLLM and SALMONN, which also use similar strategies. The proposed distillation approach, which relies on self-supervised and cross-modal transfer methods, does not go beyond well-established techniques and lacks meaningful innovation, limiting its impact.
- Unclear Model Outputs and Evaluation: The paper does not clarify how speech (waveform) is generated from the model output, leaving a gap in understanding how the system produces audible speech. There is no direct evaluation of speech quality, such as MOS (Mean Opinion Score) experiments, which would be valuable to assess the quality of the generated speech. The speaker similarity and translation quality (ASR-BLEU) metrics are missing for the speech-to-speech translation task. These would help evaluate the overall quality of the translations and speaker consistency. Besides, The evaluation of the voice assistant claims is unclear. The paper mentions the creation of a voice assistant but does not explain or evaluate what this entails, raising questions about the paper's practical applications and performance in real-world tasks.
- Lack of Comprehensive Comparisons: The paper does not compare the method with important works, such as those using cross-entropy or other distillation methods (e.g., BLSP, AudioChatLlama), which would clarify the advantages and uniqueness of the proposed approach. A more detailed comparison with existing methods like Qwen2-Audio and the Llama-3 backbone is needed, as claims about DiVA outperforming state-of-the-art models may be misleading due to discrepancies in model architectures and backbones.
- Lastly, related work and references is not incomplete, e.g., omitting numerous relevant works in the Speech LLM and instruction-following domains. This makes it harder to understand the position of this work, and what's the trade-offs and how proposed work compared with existing approaches, to help readers better understand the method's limitations and future applicability.

**Additional Comments On Reviewer Discussion:**

Although	 some of these weakness have been improved / somewhat addressed during rebuttal session, I do feel the session is too short and I would like to see a more comprehensive modification to systematically working on these suggestions.  Also the final review scores are also below the bar. Thus I recommend the authors to re-work on these weakness and re-submitting to future conferences.

---

### Decision · Program_Chairs · 2025-01-22

Reject